# Where Are the Demographic Dividends in Sub-Saharan Africa?

Michel Garenne [1,2,3,4]

1 Department of Statistics and Population Studies, University of Western Cape, Bellville 7535, South Africa; mgarenne@hotmail.com
2 MRC/Wits Rural Public Health and Health Transitions Research Unit, School of Public Health, Faculty of Health Sciences, University of the Witwatersrand, Johannesburg 3193, South Africa
3 Institut de Recherche pour le Développement (IRD), UMI Résiliences, 93140 Bondy, France
4 FERDI, Department of Economics, Université d'Auvergne, 63000 Clermont-Ferrand, France

**Abstract:** This paper reviews the concept of the demographic dividend and the empirical evidence therefor. The demographic dividend is mainly the result of fertility decline (lower number of births, lower population growth) which translates into a population age structure with a larger work force (age 15–64) and a smaller proportion of children (age 0–14), together with initially few elderly persons (age 65+). In turn, this favors economic growth, but it also has many consequences for households and for state budgets, as well as long-term consequences for population size and the environment. The first part of this paper shows the small correlations at the national macro-economic level between dependency ratios and economic growth. The second part shows the strong correlations at the household level between levels of fertility, child mortality and modern education. The third part discusses the many other correlates of the demographic dividend. The often-cited and controversial focus of the demographic dividend on economic growth hides many other positive effects of fertility control on households, on state budgets, and, in the long-run, on societies and the environment.

**Keywords:** fertility control; demographic dividend; economic growth; population growth; household wealth; health; education; state budget; environment





## 1. Introduction

Demographic transitions, social change and economic development interacted in many ways over the past two centuries. Some of these interactions are well known; others are controversial or less documented. All these changes had numerous impacts on societies and on the environment.

These interactions are studied with theoretical models and with empirical analyses, which often lead to contradictory or unexpected results, the main reason being that these dynamics could have different and independent rationales. For instance, in the classic comparison of demographic transitions and economic development between France and England, the fertility transition occurred much earlier in France, while economic development was more advanced in England [1,2].

Theoretical models also reflect this complexity. The classic Coale–Hoover model, applied to India, assumed that fertility decline will have an effect on the age structure, reducing the dependency ratio and therefore inducing more savings and promoting economic growth, starting a virtuous cycle [3,4]. In contrast, other authors argue that population pressure induces technical change (more efficient technology, higher income) and institutional change (better organization), with numerous social and demographic consequences, starting another virtuous cycle [5–15]. In the classic analysis of the "East Asian miracle", the situation is further complicated by economic policies: as dependency ratios fell, the state invested in human capital and in job creation, thereby rapidly accelerating rates of economic growth [16]. Most of those models assume implicitly that population parameters (fertility, mortality, population growth, age structure) and economic development (economic growth,

income per capita) operate in the same loops, where lower population parameters are conducive to faster economic development and where economic development is conducive to a favorable evolution of population parameters. In particular, a negative correlation is expected between the dependency ratio and economic growth. However, some of these changes can develop in another direction for independent reasons. For instance, in France, between 1900 and 2010, periods with a high dependency ratio (1960–1980) were also periods of high economic growth, whereas periods with a low dependency ratio (1900–1920 and 1930–1950) were periods of slower economic growth, partly because of the wars and partly because of the 1929 recession. As a result, in France, the correlation between dependency ratio and economic growth over a century was positive ($\rho = +0.42$) instead of negative (author's calculations). In brief, these interactions between population and development are complex and could work either way depending on the dynamics of each parameter.

The concept of demographic dividend was introduced in the literature about half a century ago [13,17–19]. It became fashionable in the early 2000s, and received much publicity from international organizations, such as UNFPA (United Nations Fund for Population Activities) and the World Bank, with the aim of promoting family planning in African countries [4,20–27]. The idea behind this approach was to say that lower fertility will lead to a lower dependency ratio and therefore to higher economic growth, producing a "dividend", like a financial investment. Even though this relationship is rational and theoretically sound, this approach appears naïve for numerous reasons, and in particular because economic growth is determined by many parameters other than age structure and savings [28,29]. More importantly, this approach ignores the many other positive consequences of fertility decline on households, on state budgets and on the environment.

The aims of this study were, firstly, to review the empirical evidence of the relationship between dependency ratio and economic growth at the national (macro) level in sub-Saharan Africa; secondly, to present the empirical evidence at the household (micro) level of the relationship between fertility level and social indicators (health and education) and, thirdly, to evoke the many other positive consequences of fertility control for state management and for the environment. Since these three parts require different approaches and different data, they are presented independently.

## 2. Dependency Ratio and Economic Growth at the Macro Level in Africa

The concept of demographic dividend at the macro level is usually presented as a virtuous cycle in which fertility decline has an impact on the age structure, changing the dependency ratio, which should lead to more savings and, therefore, more investments and more economic growth. (Figure 1).

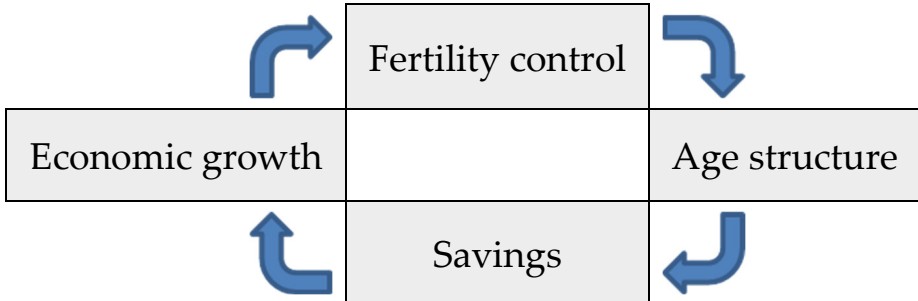

**Figure 1.** Virtuous cycle at the country (macro) level.

The key parameter of this relationship is the "dependency ratio"—that is, the ratio of the non-active population (consumers only) to the active population (producers and consumers). The lower the dependency ratio, the higher the expected economic advantage (more producers per consumer). The dependency ratio is usually measured by the ratio of the population aged 0–14 and/or 65+ to the population aged 15–64 years (youth dependency, old-age dependency and total dependency, respectively) [24,30]. In the world,

the total dependency ratio ranges from 0.18 to 1.10 by country, with an average of 0.53 in 2015–2019. In this paper, only the total dependency ratio is considered.

The dependency ratio at the national level is determined by past fertility and mortality, and in some cases by recent international migration (young adults mainly). The age structure of African populations is now well documented, and a complete database has been available since 1950 at the United Nations Population Division. For this study, the World Population Prospects, 2019 revision (WPP-2019) was used for calculating dependency ratios. The list of countries used for this analysis appears in Appendix A. The analysis covers the 1950–2019 period, presented in 5-year periods.

## 2.1. Dependency Ratios and Economic Growth in Africa

For the 46 sub-Saharan African countries as a whole, dependency ratios increased from 1950 to 1985 following mortality decline and fertility increase, then decreased after 1985, and remained high compared with other countries, averaging 0.80 in the recent period (2015–2019), compared with 0.53 in the world. The decline after 1985 was due mainly to the effects of family planning programs, which started around 1960 in Africa, first slowly, then increasingly over time, leading to a decline in total fertility rate (TFR) from 6.78 in 1975–1979 to 4.72 in 2015–2019 [31] (Figure 2).

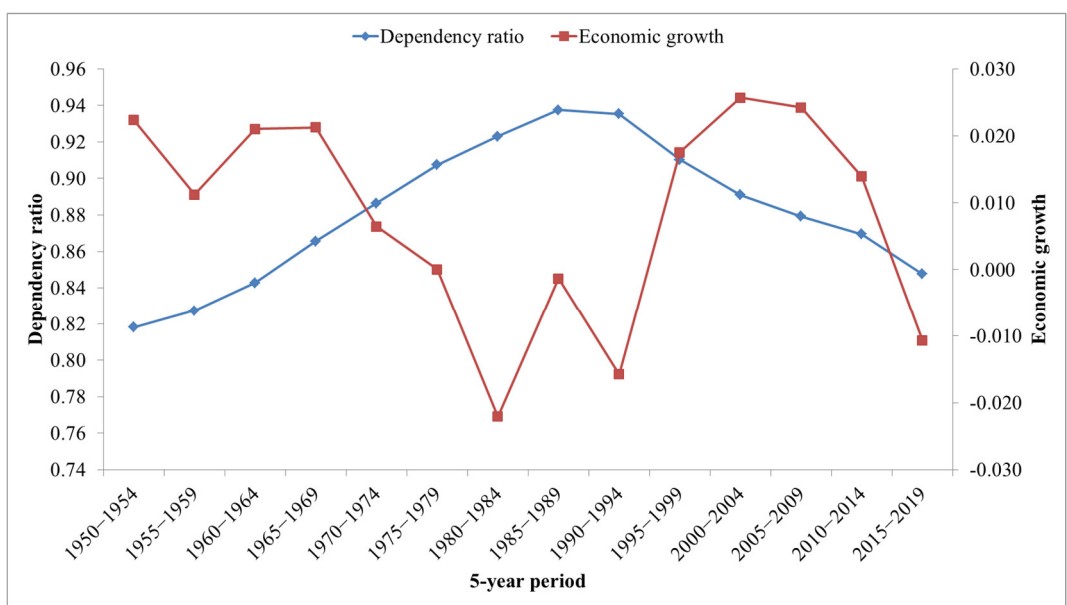

**Figure 2.** Dynamics of dependency ratio and economic growth, sub-Saharan Africa, 1950–2019. Sources: Dependency ratio: WPP-2019; economic growth: Maddison, 2010 and WDI-2022. Note: Dependency ratio is defined as numbers of people younger than 15 and older than 65 divided by numbers of people aged 15–64 years. Economic growth is defined as the average annual growth rate of GDP per capita over 5-year periods.

Economic growth in Africa is determined by a variety of factors: price of export commodities on international markets (oil, gas, minerals, agricultural products, etc.), foreign private investments, transfers from the migrant community, international public and private aid, and, of course, economic policies. This explains why economic growth behaves largely independently from the age structure and why economic growth had different dynamics from those of the dependency ratio in Africa (Figure 2).

Data for this analysis were taken from the OECD database for the 1950–1989 period [32] and from the World Bank Development Indicators [33] for the 1990–2019 period. For the continent as a whole, economic growth was stable at around 2% per year in the last decade of the colonial period (1950–1959) and the first following decade (1960–1969). Economic growth plummeted in the 1970s, mainly as a result of the oil shocks (1973, 1979) and leading to high interest rates on national debts and to mismanagement of post-colonial economies,

and became negative in the 1980s and early 1990s. Economic growth became positive again after 1995 following structural adjustment policies and various economic changes and a more favorable international environment; it then increased steadily and peaked in 2000–2009, before declining and becoming negative again in 2015–2019. Of course, these dynamics, due to economic policies and fluctuating international opportunities, in particular the price of export commodities, have little to do with the dependency ratios. Over the 1950–2019 period, the correlation by year between dependency ratio and economic growth was negative for the continent as a whole (−0.50), as expected in demographic dividend theory, but mainly for casual reasons: the post-colonial period experienced, at the same time, high fertility, a high dependency ratio and low economic growth, whereas the following period saw a small decline in the dependency ratio (due to family planning) and higher economic growth, although not in the latest period. The only common factor between these two phenomena was the intervention of the international community, which promoted, at the same time, family planning and new economic policies.

### 2.2. Dependency Ratio and Economic Growth at the Country Level

Differences by country were large in 2015–2019. Among the 46 countries, the dependency ratio ranged from 0.40 in Mauritius (the country with lowest fertility) to 1.15 in Niger (the country with highest fertility), with a standard deviation of 0.12 (Table 1).

**Table 1.** Basic values for dependency ratio and economic growth, country level, 5-year period, 46 countries in sub-Saharan Africa, 1950–2019.

| | Average | St.dev | Min | Max |
|---|---|---|---|---|
| Dependency ratio | 0.873 | 0.124 | 0.397 | 1.148 |
| Economic growth rate | +0.011 | 0.034 | −0.112 | +0.367 |
| Correlation DR/EG | −0.188 | 0.358 | −0.796 | +0.698 |

Sources: Dependency ratio = WPP-2019; economic growth: Maddison (2010) and World Bank (WDI-2022). Economic growth rates are annual rates of growth of GDP per capita over 5-year periods.

At the country level, the relationships between dependency ratio and economic growth were complex and could go in opposite directions, from strongly negative (as expected) to strongly positive over the 1950–2019 period (Tables 1 and A1, Figure 3).

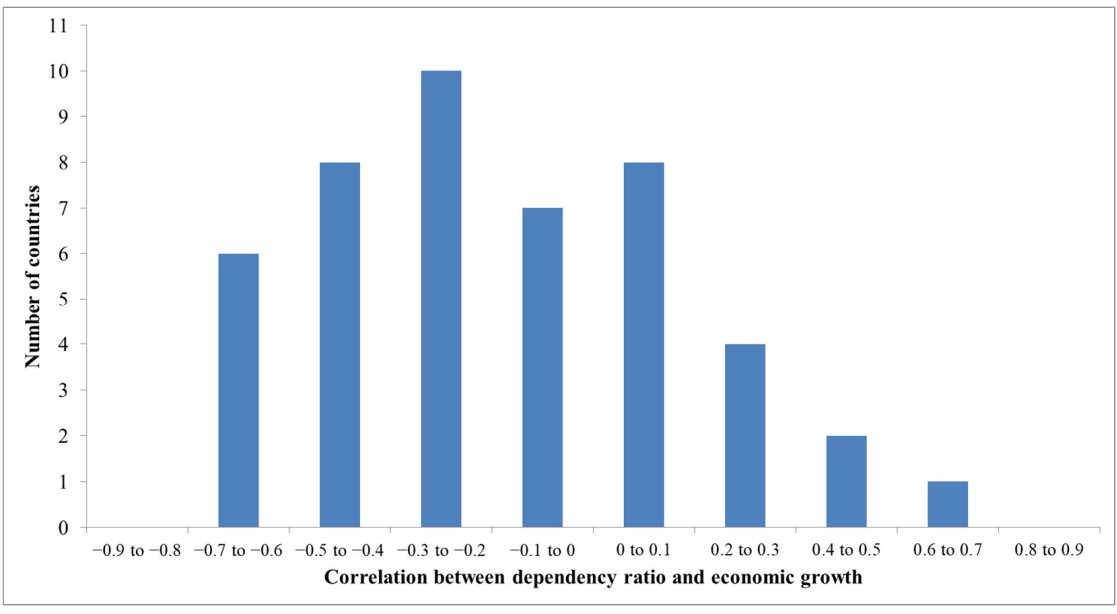

**Figure 3.** Distribution by country of correlation coefficients between dependency ratio and economic growth over the 1950–2019 period.

Correlation coefficients between dependency ratio and rate of economic growth calculated by 5-year period ranged from −0.796 (Tanzania) to +0.698 (Botswana). Some of these correlations are easy to understand, while others are unexpected or are simply spurious. For instance, in Tanzania, the dependency ratio varied very little from 1950 to 2019 (range 0.909 to 0.967) and is therefore unlikely to explain any change in economic growth rates (range −0.025 to +0.040); although the correlation between both variables was highly negative (−0.796), it is unlikely to be causal. In Botswana, the correlation (+0.698) also appeared spurious, but was easy to understand: the 1950–1974 period saw rising economic growth (due to commodity export development) and an increasing dependency ratio (due to increasing infant survival), while the following period saw declining economic growth and a declining dependency ratio (due to family planning), the two dynamics being largely independent although occurring at the same time.

A similar situation of a positive correlation could be seen in several Southern African countries (those in which periods of declining dependency ratios due to fertility decline occurred at a time of slower economic growth: South Africa, Lesotho, Zimbabwe, Seychelles), but not in others (Namibia, Swaziland, Zambia). This positive correlation also occurred in Sahelian Africa for totally different reasons (Mali, Burkina-Faso, Chad, Niger, Somalia), but not in Ethiopia, a country with strong economic growth in recent years.

Situations of a negative correlation occurred in several West-African countries (Ghana, Sierra Leone, Côte d'Ivoire, Togo, Gambia, Senegal, Guinea, Nigeria) and in several Central-African countries (Gabon, Congo, Burundi, Rwanda, Zambia, Malawi, Cameroon), as well as in others (Comoros, Madagascar, Mauritius, Sao Tome and Principe), here again mostly for independent reasons.

In conclusion, the dynamics of dependency ratio and of economic growth rate over the 1950–2019 period in Africa appear largely independent from a statistical point of view, and when they are apparently correlated it is because they occurred at the same time for independent reasons. To this observation one could add that only few African countries achieved a total fertility rate (TFR) of below 2.6 over the period, a threshold considered by UNFPA as necessary for benefiting from a favorable dependency ratio.

## 3. Demographic Dividend at the Household Level

For households (the micro level), lower fertility implies a change in household composition, fewer children to care for, a change from quantity (number of children) to quality (better care, higher education), more investments in children, higher productivity and, therefore, more wealth in the long run. This section explores the dividends at the household level in Africa. Having fewer children has many positive consequences for a household, in particular higher income per capita, and potentially more wealth, better health and better education for children. In turn, a better situation in income, health and education is conducive to lower fertility preferences. This virtuous cycle is summarized in Figure 4:

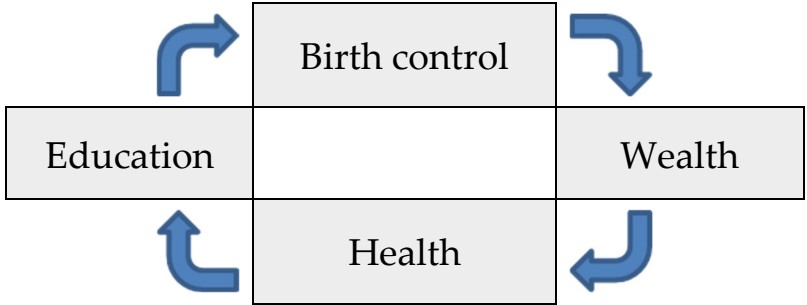

**Figure 4.** Virtuous cycle at the household level.

*3.1. Empirical Analysis*

This positive loop was verified using African Demographic and Health surveys (DHS). All African DHS with information on fertility, household wealth, child survival and level

of education were considered [34]. A total of 171 surveys were utilized, from 39 countries located in sub-Saharan Africa. Women aged 40–49 years, who were nearing the end of their reproductive life, were selected. Fertility was measured by the number of children ever born (parity), ranging from 1 to 15. Infertile women were excluded (3.0% of women), as well as the few women with 16 or more births (0.1% of women). Parity was lumped into five groups measuring the degree of fertility control: low fertility (1–3 children ever born), medium (4–6), high (7–9), very high (10–12), exceptionally high (13–15). Household wealth was measured by an absolute wealth index (AWI) counting the number of modern goods and amenities in the household. This index ranges from 0 to 16 and has been described in detail in previous studies and found to be well correlated with a large number of demographic, economic and anthropometric variables [35,36]. Health was measured by child survival, calculated simply by the ratio of children surviving over children ever born. Education of the next generation, measuring the investment of the household into children's education, was assessed by the average years of schooling of adolescents aged 15–19 in the same household as their mothers aged 40–49. For household wealth and health, the sample totaled 277,909 women. For adolescent level of education, the sub-sample totaled 201,430 adolescents and their mother. In both cases the samples were large enough to ensure statistical significance and stable relationships.

### 3.2. Household Wealth

The gradient of modern wealth by parity was smooth and pronounced: the wealth index ranged from 5.45 (parity 1–3), 4.39 (parity 4–6), 3.21 (parity 7–9), 2.72 (parity 10–12), to 2.60 (parity 13–15), a ratio of 1 to 2.09 from highest to lowest category. Women with low parity were living in much wealthier households than any other category, and the higher the parity, the lower the household wealth. Women with 10 children or more were living in households with hardly any modern equipment, whereas women with 3 children or fewer were living in households with about half of basic modern goods and amenities, and many had enough to live in a comfortable home (Figure 5).

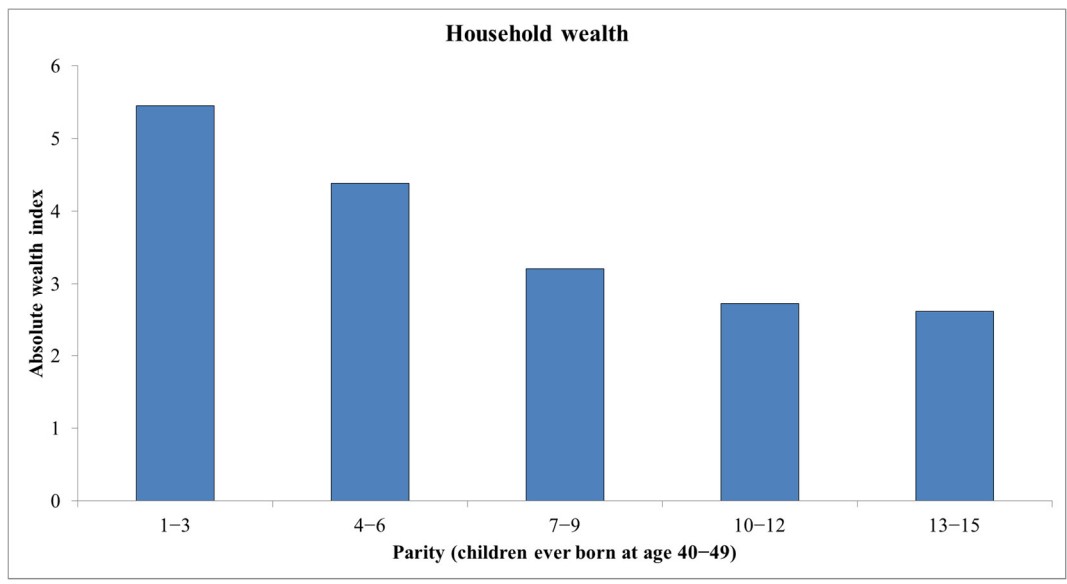

**Figure 5.** Relationship between parity and household wealth, African DHS, women aged 40–49.

### 3.3. Child Survival

Child survival is a widely used indicator of household health and correlates with most indicators of development. Its relationship with parity was also gradual and strong. Women with low parity had lost only 12.0% of their children, women with medium parity had lost 12.9% of their children, women with high parity had lost 18.4%, women with very high parity had lost 25.2% and women with extremely high parity had lost 30.7% of their children, a gradient of 1 to 2.55 from lowest to highest. Even though this relationship hides

some reverse causality, because in natural fertility situations a child loss shortly after birth implies shorter birth intervals and ultimately higher fertility, the gradient found was far stronger than that induced by early child mortality (Figure 6).

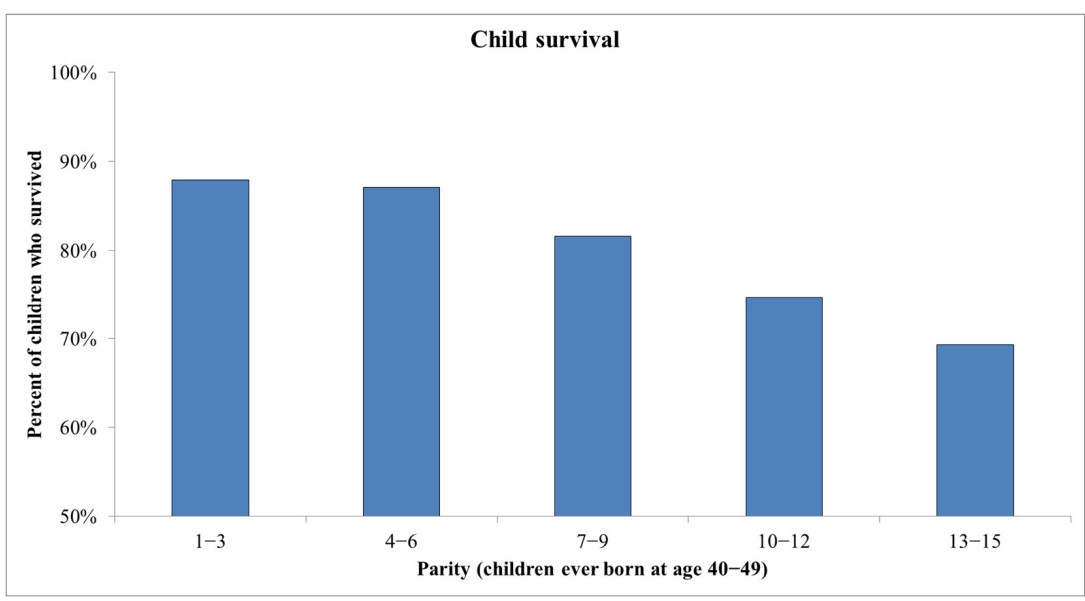

**Figure 6.** Relationship between parity and child survival, African DHS, women aged 40–49.

### 3.4. Adolescent Level of Education

The level of education of adolescents aged 15–19 was also strongly correlated with the parity of their mother aged 40–49. Adolescents whose mother had only 1–3 children completed on average 7.02 years of schooling, a much higher number of years of schooling than those whose mother had 12–15 children (4.44 years), and here again the gradient was marked and regular: 6.48 years schooling for parity 4–6, 5.23 years for parity 7–9 and 4.55 years for parity 10–12 (Figure 7). A low number of years of schooling corresponds to many children who did not even complete primary school, while 7 years ensures that most adolescents know how to read and write, and that many manage to complete several years in high (secondary) school. The differences between a high and a low level of education have many long-term consequences for households and individuals.

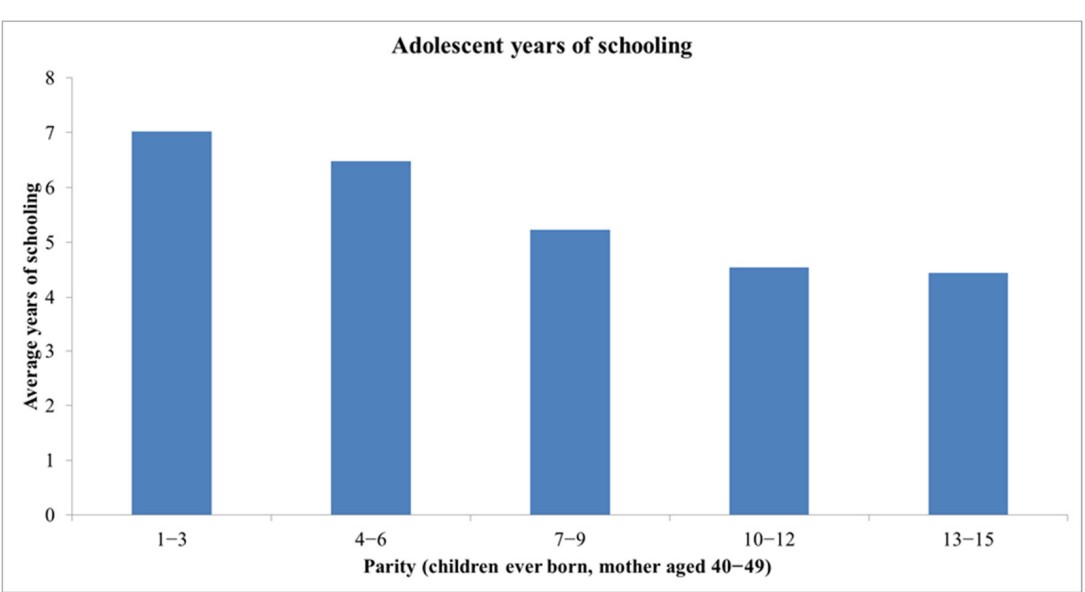

**Figure 7.** Relationship between parity and adolescent (aged 15–19) level of education, African DHS, women aged 40–49.

*3.5. Conclusion on Household Dividends*

In comparison with the lack of empirical evidence of a correlation between demographic dividend and economic growth due to the different rationales of both variables, the evidence of a correlation between fertility control and health, wealth and education at the household level was clear and in the expected direction. In this case, fertility control, household wealth, child health and adolescent level of education responded to the same rationales and the same logic of development and exhibited the same correlations between variables.

## 4. Other Aspects of the Demographic Dividend

The first two sections of this study briefly reviewed the demographic dividend at the macro (country) and micro (household) levels. There are many other positive outcomes of lower fertility and lower dependency ratios for the economy, for state budgets, for peace and security, and for the environment. It is beyond the scope of this paper to explore all the positive consequences of lower fertility, and only some of them are described in this section.

*4.1. Entry into the Labor Force and the "Youth Bulge"*

Another consequence of a lower dependency ratio due to fertility decline is a lower risk of the negative consequences of the "youth bulge"—that is, a very high proportion of young adults, say age 15–24. Firstly, reducing this proportion facilitates the entry of young adults into the labor force and their integration in society [37]. Secondly, it reduces the risk of conflicts associated with the youth bulge [38–40].

*4.2. State Budgets*

For state management, a lower dependency ratio due to lower fertility implies fewer social investments related to children, in particular in schools and services for maternal and child health. Such a saving could be used for more productive investments by the state (in agriculture, industry, transport and other services) and more protective investments (security, environment). This, in turn, could induce more economic growth and, even more importantly, more ecologic growth, more sustainable development, respecting and protecting the environment.

*4.3. GDP per Capita*

In some African countries, population growth rate often exceeded the GDP growth rate, therefore turning the GDP per capita growth rate into negative values, which has numerous economic and social consequences. In the database of 46 African countries between 1950 and 2019, 30.4% of the 5-year periods were affected by negative economic growth per capita. Reducing fertility will have an impact on population growth and, therefore, on GDP per capita. Rapid population growth also has numerous other consequences [41].

*4.4. Environment*

Another consequence of lower fertility is slower population growth, which has numerous serious and long-term consequences on the environment. Although this is not a direct effect of the dependency ratio, it has the same underlying factor: the fertility decline.

Population growth has been recognized for a long time as a major issue for equilibrium between population and environment, going back at least to the days of Thomas Malthus [42]. One issue is the amount of arable land per capita, the main factor of food production. A second issue is water resources per capita, which is becoming a key issue in many countries and a source of conflicts. A third issue is that excess population may induce forced migration, which pose numerous problems for peace and stability.

Population growth and population size also have a major impact on global warming. The IPCC (Intergovernmental Panel on Climate Change) considers that high rates of population growth, population size and population density are the main factors of global climate change [43,44]. The emission of greenhouse-effect gases is in fact proportionate

to population size. These gases induce global warming, with endless environmental consequences. Furthermore, population size and density also have many consequences for levels of pollution (air, water, soil) and for the transmission of communicable diseases, such as COVID-19. Global warming also has an effect on vector borne diseases, such as malaria and dengue.

## 5. Discussion and Conclusions

The literature on the demographic dividend focuses on age structure and the immediate benefits of slower population growth and of a lower dependency ratio. However, this focus should not be allowed to hide the other consequence in the long run, which is population aging. In several countries of Europe and East Asia (such as China, Korea and Japan), the higher dependency ratio due to very low fertility and a high proportion of elderly people in the population is seen as "the other population problem" [45]. This trend is becoming a major concern in many countries, in particular as regards old-age support, services for the elderly, age at retirement and the financing of old-age pensions.

A lower dependency ratio could also be due to factors other than fertility decline, in particular the immigration of young adults, as was previously the case in the USA and is now the case in the Gulf states. In this case, the decline in the dependency ratio is due to an increase in the adult working population and not to a proportionate decline in the child population. In this case, too, the correlation with income growth is likely to be high, because it is due to reverse causality: high economic growth induces high demand for manpower, increasing the immigration of young workers and decreasing the dependency ratio.

The often-quoted impact of the demographic dividend on economic growth appears illusory because economic growth responds primarily to other rationales. The components of economic growth are complex, since they involve not only savings (at national level) but also other measurable components, such as foreign investments, transfers and exports of natural resources (energy, minerals), as well as non-measurable components, such as innovation, governance and the efficiency of the banking system. As a result, the correlation between age structure and economic growth appears to be buried among the many other factors of economic growth. In order to prove or measure an effect of the demographic dividend one should take into account all the factors of economic growth, which is difficult to do, if not impossible. In fact, in Africa, many of the factors of economic growth are external, not to mention that their effects depend also on internal management of the economy, which can be erratic. However, it should be recognized that a lower dependency ratio is expected to have positive effects on economic growth, even if they cannot be measured.

This study showed many correlates between lower fertility and household parameters. This point should be better documented and displayed and is probably a better argument to promote family planning than economic growth. Only a few parameters were considered in this study, but there are many others, in particular the nutritional status of children. Furthermore, as shown by the European experience, these changes cumulate over time and have long-term intergenerational effects on family structure, fertility preferences, family relationships and inheritance, as well as endless consequences in the long run.

The effects of lower fertility on state budgets are better known, but they assume that the benefits are properly used for public benefit, and not wasted on useless expenses or the aggrandizement and enrichment of elites.

Lastly, the main effect of fertility decline is slower population growth, which should lead, it is hoped, toward zero or negative population growth, necessary to establish a new equilibrium between population and the environment at global level [46,47]. The demographic situation of sub-Saharan African countries is still far from this ideal, and much remains to be done to reduce population growth in order to get closer to it.

**Funding:** This research received no external funding.

**Institutional Review Board Statement:** Not applicable.

**Informed Consent Statement:** Not applicable.

**Data Availability Statement:** All data are in public access.

**Acknowledgments:** The author thanks all institutions who provided open access to statistical databases, in particular the United Nations Population Division, the World Bank, OECD and the DHS program. The author also thanks the editor and the three reviewers for useful and constructive comments.

**Conflicts of Interest:** The author declares no conflict of interest.

## Appendix A

**Table A1.** List of countries used for the empirical analysis of correlations by 5-year period, 1950–2019.

| Country | Dependency Ratio | Economic Growth | Correlation |
|---|---|---|---|
| Angola | 0.948 | −0.002 | −0.166 |
| Benin | 0.877 | 0.007 | +0.239 |
| Botswana | 0.854 | 0.040 | +0.698 |
| Burkina Faso | 0.900 | 0.018 | +0.046 |
| Burundi | 0.933 | 0.003 | −0.500 |
| Cameroon | 0.879 | 0.010 | −0.385 |
| Cape Verde | 0.877 | 0.032 | +0.364 |
| Central African Republic | 0.812 | −0.006 | −0.320 |
| Chad | 0.941 | 0.005 | +0.162 |
| Comoros | 0.854 | 0.003 | −0.545 |
| Congo (RDC) | 0.912 | −0.009 | −0.201 |
| Congo (Rep.) | 0.862 | 0.004 | −0.014 |
| Côte d'Ivoire | 0.845 | 0.006 | −0.679 |
| Djibouti | 0.837 | 0.004 | −0.582 |
| Equatorial Guinea | 0.779 | 0.055 | +0.218 |
| Ethiopia | 0.915 | 0.021 | −0.163 |
| Gabon | 0.758 | 0.003 | −0.611 |
| Gambia | 0.863 | 0.005 | −0.447 |
| Ghana | 0.857 | 0.011 | −0.775 |
| Guinea | 0.828 | 0.016 | −0.352 |
| Guinea-Bissau | 0.846 | 0.013 | −0.277 |
| Kenya | 0.971 | 0.011 | −0.130 |
| Lesotho | 0.863 | 0.025 | +0.465 |
| Liberia | 0.856 | −0.004 | −0.197 |
| Madagascar | 0.893 | −0.006 | −0.560 |
| Malawi | 0.951 | 0.014 | −0.402 |
| Mali | 0.880 | 0.013 | +0.035 |
| Mauritania | 0.869 | 0.012 | −0.235 |
| Mauritius | 0.666 | 0.028 | −0.291 |
| Mozambique | 0.885 | 0.013 | +0.067 |
| Namibia | 0.831 | 0.011 | −0.164 |
| Niger | 1.008 | −0.001 | +0.193 |
| Nigeria | 0.857 | 0.011 | −0.350 |

**Table A1.** *Cont.*

| Country | Dependency Ratio | Economic Growth | Correlation |
|---|---|---|---|
| Rwanda | 0.962 | 0.018 | −0.424 |
| Sao Tome and Principe | 0.887 | 0.016 | −0.336 |
| Senegal | 0.908 | 0.004 | −0.357 |
| Seychelles | 0.693 | 0.022 | +0.255 |
| Sierra Leone | 0.802 | 0.008 | −0.691 |
| Somalia | 0.909 | 0.004 | +0.107 |
| South Africa | 0.712 | 0.008 | +0.100 |
| Swaziland | 0.926 | 0.025 | −0.123 |
| Tanzania | 0.935 | 0.013 | −0.796 |
| Togo | 0.892 | 0.007 | −0.677 |
| Uganda | 0.998 | 0.010 | +0.409 |
| Zambia | 0.951 | 0.009 | −0.428 |
| Zimbabwe | 0.930 | 0.006 | +0.189 |

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
