# Peer review of "Where Are the Demographic Dividends in Sub-Saharan Africa?"

_world, doi:10.3390/world4030038_

Round 1

Reviewer 1 Report

The manuscript presents an interesting subject, the demographic dividend in Sub-Saharan countries and provides and exhaustive and well structured approach using various relevant data sources. I particularly enjoyed challenging the usual view that simplifies economic expansion through a simple formula of increase in working age population, albeit being a pivotal factor, can't be alone explain wealth.

There are a few issues that I have found in the manuscript.

1. One issue that makes reading difficult is the often colloquial style of the manuscript.

An example: 'more productive adults and less unproductive children and elderly persons' , where 'productive adults' should be 'working force population'. The 'less productive' part of the population (children and elderly) is debatable as the share of children who 'study and work' or 'work only' in low and middle income countries (such as in Sub-Saharan countries) is the highest in the world (see World Bank statistics for reference). 

2. Another issue that I have noticed early on and that I have found quite surprising is the definition of population dividend. At the beginning of the manuscript the PD is defined as solely based on decreasing fertility, whereas it should take into account the dynamics between three age groups over time. (see UNFPA definition below)

the economic growth potential that can result from shifts in a population’s age structure, mainly when the share of the working-age population is larger than the non-working-age share of the population 

which usually involves a decline in mortality, the subsequent increase in all age groups and lastly a decrease in fertility.

3. The discussion introduces the concept of zero growth as ideal and desirable, without presenting supporting literature. It also mentions that zero growth can help produce an equilibrium between the population and the environment, also without introducing or providing supporting literature previously.

4. I was surprised to see section 3 providing a description of positive effects of the PD right after the results section. It could be integrated in the results section or in the introduction or in the discussion, since it does not seem to be based on data analysis but on literature.

The text could benefit from a minor English revision, as some expressions and definitions seem to be off 

e.g. Far-East Asia = East Asia like Japan and South Korea or South East Asia like Vietnam)

The other demographic dividends other selected demographic dividend definitions

Author Response

Reviewer 1

The manuscript presents an interesting subject, the demographic dividend in Sub-Saharan countries and provides and exhaustive and well structured approach using various relevant data sources. I particularly enjoyed challenging the usual view that simplifies economic expansion through a simple formula of increase in working age population, albeit being a pivotal factor, can't be alone explain wealth.

There are a few issues that I have found in the manuscript.

  1. One issue that makes reading difficult is the often colloquial style of the manuscript. An example: 'more productive adults and less unproductive children and elderly persons' , where 'productive adults' should be 'working force population'. The 'less productive' part of the population (children and elderly) is debatable as the share of children who 'study and work' or 'work only' in low and middle income countries (such as in Sub-Saharan countries) is the highest in the world (see World Bank statistics for reference). 

            OK, change made.

  1. Another issue that I have noticed early on and that I have found quite surprising is the definition of population dividend. At the beginning of the manuscript the PD is defined as solely based on decreasing fertility, whereas it should take into account the dynamics between three age groups over time. (see UNFPA definition below): “the economic growth potential that can result from shifts in a population’s age structure, mainly when the share of the working-age population is larger than the non-working-age share of the population which usually involves a decline in mortality, the subsequent increase in all age groups and lastly a decrease in fertility”.

            We disagree, and show the opposite. When mortality declines, the size of the 0-14 age group increases, which increases the dependency ratio (as shown in Figure 2). The only factors of declining dependency ratio are the fertility decline and the immigration of young adults (work force). This is well mentioned in the paper. This should not be confused with the demographic transition, which involves a mortality decline followed by a fertility decline.

  1. The discussion introduces the concept of zero growth as ideal and desirable, without presenting supporting literature. It also mentions that zero growth can help produce an equilibrium between the population and the environment, also without introducing or providing supporting literature previously.

            Good point. A reference is now added.

  1. I was surprised to see section 3 providing a description of positive effects of the PD right after the results section. It could be integrated in the results section or in the introduction or in the discussion, since it does not seem to be based on data analysis but on literature.

            We disagree. This is part of the discussion, and addresses other issues than those shown in the paper.

Comments on the Quality of English Language

The text could benefit from a minor English revision, as some expressions and definitions seem to be off : e.g. Far-East Asia = East Asia like Japan and South Korea or South East Asia like Vietnam)

            OK, done.

Reviewer 2 Report

This paper is a very welcome addition to the literature, in that it dispels some myths about a “demographic dividend” of increased economic growth flowing automatically from decreased fertility levels, and highlights instead the very real (and automatic) benefits of reduced fertility for households and societies.

A couple of suggestions for the author:

1.     In the introduction, it could be made clearer that the concept of the “demographic dividend” goes further than the more mechanistic argument that Coale and Hoover developed in the 1950s (fewer children, more savings). The term “demographic dividend” as built around the “East Asian Miracle” described a situation where, as dependency ratios fell,  the state invested in human capital and in job creation, thereby rapidly accelerating rates of economic growth. Since then, the term has come to be used much more loosely, more akin to the Coale-Hoover concept and disassociated from growth in human capital and in jobs.

2.     In Figures 5-7, it would be helpful to add a note specifying the controls used in the analysis. This would help the reader understand that it is not just a matter of wealthier /more educated households having fewer children and investing more in them --- but that less-endowed households also improve their situation by having fewer children as indicated in Figure 4.

3.     This is not directly relevant to the central argument of the paper, but the author could also mention briefly the growing concern that without increased fertility there will be fewer working age to support the aged. This is essentially arguing for a Ponzi scheme whereby the world's population will continue to grow indefinitely in order to support (growing) cohorts of the elderly --- a situation clearly not sustainable for the planet.

Author Response

Reviewer 2

This paper is a very welcome addition to the literature, in that it dispels some myths about a “demographic dividend” of increased economic growth flowing automatically from decreased fertility levels, and highlights instead the very real (and automatic) benefits of reduced fertility for households and societies.

A couple of suggestions for the author:

  1. In the introduction, it could be made clearer that the concept of the “demographic dividend” goes further than the more mechanistic argument that Coale and Hoover developed in the 1950s (fewer children, more savings). The term “demographic dividend” as built around the “East Asian Miracle” described a situation where, as dependency ratios fell, the state invested in human capital and in job creation, thereby rapidly accelerating rates of economic growth. Since then, the term has come to be used much more loosely, more akin to the Coale-Hoover concept and disassociated from growth in human capital and in jobs.

            OK, a sentence was added.

  1. In Figures 5-7, it would be helpful to add a note specifying the controls used in the analysis. This would help the reader understand that it is not just a matter of wealthier /more educated households having fewer children and investing more in them --- but that less-endowed households also improve their situation by having fewer children as indicated in Figure 4.

            These are plain data. There are no controls added.

  1. This is not directly relevant to the central argument of the paper, but the author could also mention briefly the growing concern that without increased fertility there will be fewer working age to support the aged. This is essentially arguing for a Ponzi scheme whereby the world's population will continue to grow indefinitely in order to support (growing) cohorts of the elderly --- a situation clearly not sustainable for the planet.

            Good point. A sentence was added in the discussion.

Reviewer 3 Report

This paper mainly discusses the phenomenon of Demographic dividend in Sub-Saharan Africa, and attempts to analyze the impact of demographic age structure changes on economic growth from both macro and micro perspectives.

The article has clear ideas and relatively simple methods. The following suggestions are mainly proposed:

1. In the process of age structure changes, not only the change in fertility rate should be considered, but also the increase in population life expectancy. The author needs to add this section.

2. The impact of education on fertility needs further in-depth analysis.

3. Is there a situation where fertility first decreases and then increases?

4. The discussion section needs to be more in-depth.

no

Author Response

Reviewer 3

This paper mainly discusses the phenomenon of Demographic dividend in Sub-Saharan Africa, and attempts to analyze the impact of demographic age structure changes on economic growth from both macro and micro perspectives.

The article has clear ideas and relatively simple methods. The following suggestions are mainly proposed:

  1. In the process of age structure changes, not only the change in fertility rate should be considered, but also the increase in population life expectancy. The author needs to add this section.

            This is another issue. When life expectancy increases, firstly the size of the child population increases, then the size of the elderly population increases, both increasing the dependency ratio. Both issues are addressed briefly in the paper.

  1. The impact of education on fertility needs further in-depth analysis.

            This is another issue, out of context. The author has written on this issue in a longitudinal perspective, and will be happy to say more if needed.

  1. Is there a situation where fertility first decreases and then increases?

            This is another issue. This has happened in the past in Europe and in the USA (the baby boom of 1945-1964). In Africa, only situation of “fertility stalls” were documented, and so far with a marginal effect on the dependency ratio.

  1. The discussion section needs to be more in-depth.

            We will be happy to extend the discussion if needed. But to add what?

Round 2

Reviewer 1 Report

This paper was a very interesting read and I was happy to be involved in the review process. I am very satisfied with the improved manuscript and, even if some comments were not addressed I understand it is based on different styles of structuring papers.

I look forward to seeing the paper published.

Author Response

As requested, I changed the reference system for numbers called in text. In addition, I corrected two small typos.

I hope that this version will be acceptable.

Thank you for giving me the opportunity to publish in your Journal “World”.

Kind regards,

Michel Garenne